# ARTEMIS: ATTENTION-BASED RETRIEVAL WITH TEXT-EXPLICIT MATCHING & IMPLICIT SIMILARITY

**Ginger Delmas**      **Rafael S. Rezende**      **Gabriela Csurka**      **Diane Larlus**

NAVER LABS Europe

## ABSTRACT

An intuitive way to search for images is to use queries composed of an example image and a complementary text. While the first provides rich and implicit context for the search, the latter explicitly calls for new traits, or specifies how some elements of the example image should be changed to retrieve the desired target image. Current approaches typically combine the features of each of the two elements of the query into a single representation, which can then be compared to the ones of the potential target images. Our work aims at shedding new light on the task by looking at it through the prism of two familiar and related frameworks: text-to-image and image-to-image retrieval. Taking inspiration from them, we exploit the specific relation of each query element with the targeted image and derive light-weight attention mechanisms which enable to mediate between the two complementary modalities. We validate our approach on several retrieval benchmarks, querying with images and their associated free-form text modifiers. Our method obtains state-of-the-art results without resorting to side information, multi-level features, heavy pre-training nor large architectures as in previous works. Our code is available at `https://github.com/naver/artemis`.

## 1 INTRODUCTION

When using an image search engine, should the user provide a detailed textual description or a visual example? These two image retrieval tasks, respectively studied as cross-modal search (Wang et al., 2016) and visual search (Philbin et al., 2007) in the computer vision community, have been the topic of extensive studies.

However, limiting search queries to a single modality is restrictive. On the one hand, text proposes an accurate but only partial depiction of the desired result, as it is impossible to provide an exhaustive textual description. On the other hand, visual queries are richer but a lot more ambiguous: how should the system guess which characteristics the user wants to maintain for the target image? The task of **image search with free-form text modifiers** reduces this ambiguity by allowing a dual query composed of an example (called reference) image, and a textual description (called text modifier) which explains how the reference image should be modified to obtain the desired result (the target image). This task is illustrated in Figure 1. From this formulation, a challenging research question arises: *How should the two facets of this dual query be leveraged and combined?*

The first methods to answer this question directly are borrowed from the Visual Question Answering literature (Kim et al., 2016; Santoro et al., 2017). The most standard approach for this task consists in fusing the features of the two components of the query into a single representation, so it can be compared to the representation of any potential target image (Vo et al., 2019; Chen & Bazzani, 2020). Among the current strategies, some resort to rich external information (Chen et al., 2020; Liu et al., 2021) while others rely on multi-level visual representations (Chen et al., 2020) or heavy cross-attention architectures (Hosseinzadeh & Wang, 2020; Chawla et al., 2021; Liu et al., 2021).

Departing from these strategies, we draw inspiration from two fields related to the task, cross-modal and visual search, and we advocate for a combination of the two components of the query taking into account their specific relationship with the target image. Additionally, we note that the text modifier provides some insights about what *should change* (as an explicit request from the user), and what *should be preserved* in the target image with respect to the reference image (*i.e.* what is

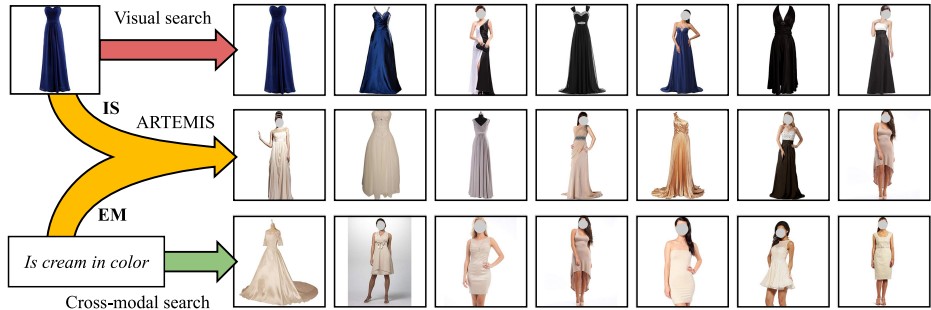

Figure 1: **Image search with free-form text modifiers** (shown in yellow) enriches the visual query with text in natural language. The task differs from visual search (using only the reference image, shown in red), and cross-modal search (using only the textual cue, shown in green). Yet, these can be seen as two complementary aspects of the studied task. Our method leverages both modalities to look for target images that i) *explicitly match* (EM) the characteristics mentioned in the text, and ii) bear some resemblance with the reference image using text-guided *implicit similarity* (IS).

not mentioned in the text modifier). All these observations lead to the design of two independent modules, one for each modality, that are eventually combined.

Our *Explicit Matching* (EM) module measures the compatibility of potential target images with the properties mentioned in the textual part of the query, as a form of cross-modal retrieval. Simultaneously, our *Implicit Similarity* (IS) module considers the relevance of the target images with respect to the properties of the reference image implied by the textual modifier.

Additionally, each of our modules uses a light-weight attention mechanism, guided by the text modifier which helps selecting the characteristics of the target image to focus on for retrieval. This results in our proposed model, named **A**ttention-based **R**etrieval with **T**ext-**E**xplicit **M**atching and **I**mplicit **S**imilarity (**ARTEMIS**), which jointly learns these two modules, their respective attention mechanisms, and the way they should be combined, in a unified manner.

In summary, our contribution is twofold. First, we suggest a new way to look at the task of image search with free-form text modifiers, where we draw inspiration from – and combine – the two fields of cross-modal and visual retrieval. Second, we propose a computationally efficient model based on two complementary modules and their respective text-guided light-weight attention layers, where each handles one modality of the query. These modules are trained jointly.

We experimentally validate ARTEMIS on several datasets and show that, despite its simplicity in terms of architecture and training strategy, it consistently outperforms the state of the art.

## 2 RELATED WORK

Image retrieval with purely visual queries has been extensively studied and is typically referred to as **instance-level image retrieval** (Philbin et al., 2007). Most approaches either learn global image representations for efficient search (Perronnin et al., 2010; Radenović et al., 2016; Gordo et al., 2016) or extract local descriptors for a costly yet robust matching (Arandjelović & Zisserman, 2012; Tolias et al., 2016; Noh et al., 2017). Current state-of-the-art models use both global and local descriptors sequentially, using the latter to refine the initial global ranking (Cao et al., 2020; Tolias et al., 2020).

The most standard retrieval scenario involving text, **cross-modal retrieval**, uses a textual description of the image of interest as a query. A common approach is to learn a joint embedding space and modality-specific encoders producing global representations in that shared space. Then, the task boils down to computing distances between those representations (Frome et al., 2013; Wang et al., 2016; Faghri et al., 2018). The most recent works focus on cross-attention between text segments and image region features (Lee et al., 2018; Li et al., 2019; 2020). However, these models do not scale, and they require to score each potential text-query target-image pair independently. Miech et al. (2021) address this issue by distilling a cross-attention architecture into a dual encoder one.

Despite similitudes between these two tasks and the one of **image search with text modifiers**, the first baselines for the latter initially adapted visual question answering (VQA) approaches (Kim et al., 2016). To reason with features from different modalities, Santoro et al. (2017) use a sequence of MLPs, whereas FiLM (Perez et al., 2018) injects text features into the image encoder at multiple layers, altering its behavior through complex modifications.

Current approaches address this task by composing the two query elements into a single joint representation that is compared to any potential target image feature. As such, in TIRG (Vo et al., 2019), the reference image and the "relative captions" (or text modifiers) are fused through a gating-residual mechanism, and the text feature acts as a bridge between the two images in the visual representation space. Anwaar et al. (2021) use an autoencoder-based model to map the reference and the target images into the same complex space and learn the text modifier representation as a transformation in this space. Lee et al. (2021) and Chawla et al. (2021) both propose to disentangle the multi-modal information into content and style. Chen & Bazzani (2020) resort to image's descriptive texts as side information to train a joint visual-semantic space, training a TIRG model on top. The alignment of visual and textual features for regularisation is reused in VAL (Chen et al., 2020) which also inserts a composite transformer at many levels of the visual encoder to preserve and transform the visual content depending on the text modifier; the model is then optimized with a hierarchical matching objective. Hosseinzadeh & Wang (2020) also align image and text through a cross-modal module, but they use region proposals instead of CNN activations. Liu et al. (2021) propose a transformer-based model that leverages rich pre-trained vision-and-language knowledge for modifying the visual features of the query conditioned by the text. We note that these last methods are reminiscent of the heavy cross-attention models and share the same limitations with respect to scaling.

In contrast to most methods described above, ARTEMIS does not compose modalities into a joint global feature for the query (Vo et al., 2019; Chen & Bazzani, 2020; Lee et al., 2021), does not compute costly cross-attention involving the target image (Hosseinzadeh & Wang, 2020; Chawla et al., 2021) and does not extract multi-level visual representations (Chen et al., 2020). Instead it leverages the textual modifier in simple attention mechanisms to weight the dimensions of the visual representation, emphasizing the characteristics on which the matching should focus. This results in a model with a manageable amount of parameters to learn, efficient inference on the query side, and that leads to state-of-the-art results.

The term **attention** has been heavily used in the past, referring to light-weighting mechanisms (Xu et al., 2015; Li et al., 2017; Kang et al., 2018; Wang et al., 2019) similarly to us, or to more complex weighting mechanisms (Lu et al., 2016). This differs from the more recent definition of *attention* introduced by Vaswani et al. (2017), which became standard in many approaches tackling the cross-modal retrieval (Lee et al., 2018; Li et al., 2019; 2020) and VQA (Bai et al., 2018) tasks.

## 3 PROPOSED METHOD

This section describes ARTEMIS, our proposed approach for the task of image search with free-form text modifiers. In this setting, queries are bimodal pairs, composed of a reference image $I_r$ and a text modifier $T_m$. Such queries are used to retrieve any relevant image $I_t$ from a gallery of images.

Let $\phi(.)$ and $\theta(.)$ be the visual and textual encoders respectively. We denote the resulting L2-normalized features of the reference image $r$, of the text modifier $m$, and of the target image $t$, *i.e.* $r = \phi(I_r) \in \mathbb{R}^{H_I}$, $m = \theta(T_m) \in \mathbb{R}^{H_T}$, and $t = \phi(I_t) \in \mathbb{R}^{H_I}$.

**Two complementary views of the task.** Both the textual and the visual cues are important to find relevant target images. Consider for example the query in Figure 1. The reference image provides strong visual context and displays some semantic details – *e.g.* . the dress' length, among other style components– that should be shared with the target image, to the extent that the text does not specify their alteration. In other words, by the absence of an explicit modification request of some properties, the modifying text implicitly "validates" the visual cues from the reference image associated to those properties, and these cues can be used to directly query for the target image. Hence, part of the task boils down to image-to-image retrieval (visual search), implicitly conditioned on the text modifier.

On the other hand, the textual cue explicitly calls for modifications of the reference image. It was even observed that, in some cases, users tend to directly describe characteristics of the target image,

without explicitly stating the discrepancy between the reference and the target image[1] (*c.f.* the analysis presented for FashionIQ in (Wu et al., 2021) and for CIRR in (Liu et al., 2021)) Therefore the text, which describes mandatory details for the target, can be directly used to query for the target image, as is common in text-to-image retrieval. Yet, due to the specificities of the task, we go beyond the usual text-to-image retrieval.

**Proposed approach.** We propose to revisit text-to-image and image-to-image retrieval to tackle the task at stake, which is at the intersection of both. Standard strategies, for both, boil down to directly comparing the global feature of the query (image or text) to those of the potential targets, producing two independent, modality-specific, and not necessarily immediately compatible scoring strategies. In particular, directly comparing the representations of the reference and the target images is inaccurate as the two images should differ, based on the modifying text. Likewise, directly comparing the representations of the modifying text and the target image is insufficient and potentially detrimental since part of the target image should stay similar to the reference image.

Here, we reconcile the two tasks and enhance them by designing a model which is trained to match i) the characteristics of the reference image to the ones of the target image and ii) the requirements provided by the text with properties of the target image. Based on the intuitions above, in both cases we use the text to design an attention mechanism that selects the visual cues which should be emphasized during matching.

The first objective is carried out by the *Implicit Similarity* module (Section 3.1), which draws from visual search, and the second one by the *Explicit Matching* module (Section 3.2), inherited from cross-modal retrieval. These two modules, jointly trained, respectively output the IS and the EM scores, which are simply summed into one, as a form of late fusion (Section 3.3). The full architecture of the model is illustrated in Figure 2.

## 3.1 IMPLICIT SIMILARITY

In the first module, the textual cue $m$ guides the comparison between the target and the reference images; it is used to define an attention mechanism $\mathcal{A}_{IS}(.) : \mathbb{R}^{H_T} \to [0, 1]^{H_I}$, taking $m$ as input, which predicts the importance of the visual representation dimensions when comparing the target to the reference image. The IS score is then computed as:

$$s_{IS}(\boldsymbol{r}, \boldsymbol{m}, \boldsymbol{t}) = \langle \mathcal{A}_{IS}(\boldsymbol{m}) \odot \boldsymbol{r} \mid \mathcal{A}_{IS}(\boldsymbol{m}) \odot \boldsymbol{t} \rangle, \tag{1}$$

where $\odot$ is the pointwise product and $\langle . | . \rangle$ represents the cosine similarity between the two L2-normalized terms. $\mathcal{A}_{IS}(.)$ is a two-layer MLP separated by a ReLU, and followed by a softmax.

## 3.2 EXPLICIT MATCHING

This second module captures how well the target image matches the textual cue. As the target image $t$ and the textual modifier $m$ belong to different modalities, we first project $m$ into the visual space with a learned linear transformation $\mathcal{T}(.) : \mathbb{R}^{H_T} \to \mathbb{R}^{H_I}$. In addition, since the textual modifier should not act on all the characteristics of the image ( the target image should remain partially similar to the reference image), we rely on a second attention mechanism, $\mathcal{A}_{EM}(.) : \mathbb{R}^{H_T} \to [0, 1]^{H_I}$ to emphasize the dimensions of the image feature to focus on for matching.

Hence, the EM score is computed as:

$$s_{EM}(\boldsymbol{m}, \boldsymbol{t}) = \langle \mathcal{T}(\boldsymbol{m}) \mid \mathcal{A}_{EM}(\boldsymbol{m}) \odot \boldsymbol{t} \rangle. \tag{2}$$

where $\mathcal{T}(.)$ consists in a fully connected layer. Note that $\mathcal{A}_{IS}(.)$ and $\mathcal{A}_{EM}(.)$ share the same architecture but not their weights, as the role they give to $m$ is different. Indeed, intuitively $\mathcal{A}_{EM}(.)$ focuses on what is mentioned by the textual modifier while $\mathcal{A}_{IS}(.)$ looks for what is not. Put differently, $\mathcal{A}_{EM}(.)$ "up-weights" the dimensions in the image feature that correspond to the semantic information provided by the textual cue, whereas $\mathcal{A}_{IS}(.)$ "down-weights" these dimensions to rather focus on the shared characteristics between the reference and target images.

---

[1]Not having the reference image *explicitly* referred to in the text does not mean it is useless (*c.f.* first view).

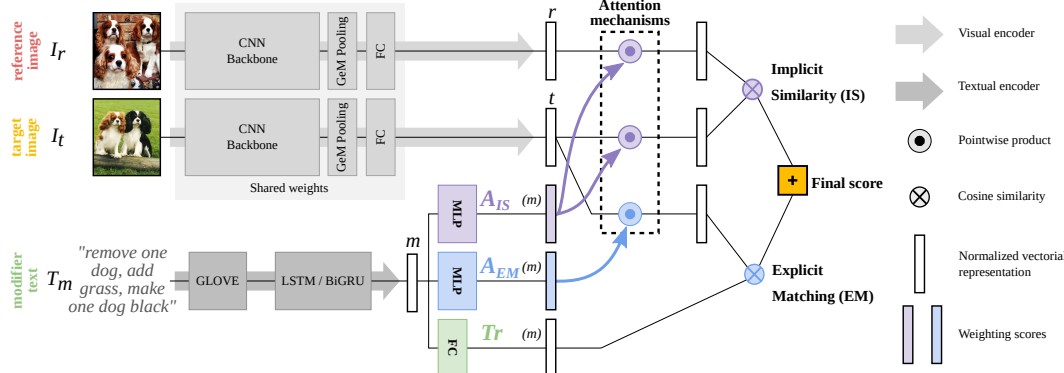

Figure 2: **Illustration of the proposed architecture.** Our model takes a triplet (reference image $I_r$, target image $I_t$, text modifier $T_m$) as input and feed each element to its respective encoder. The text feature is then used in multiple ways (including two attention mechanisms) to compute the implicit similarity (IS) and the explicit matching (EM) scores. These two complementary retrieval scores are summed to produce the final score which is used to rank potential target images.

### 3.3 TRAINING PIPELINE

Given a minibatch of $B$ training samples $\mathbb{B} = \{(I_{r_i}, T_{m_i}, I_{t_i})\}_{i=1}^{B}$ where $I_{t_i}$ corresponds to an image that is annotated as a correct target for the query $(I_{r_i}, T_{m_i})$, we train our model using the batch-based classification (BBC) loss:

$$\mathcal{L}_{BBC}(\mathbb{B}) = -\frac{1}{B} \sum_{i=1}^{B} \log \frac{\exp\{\gamma \, s(\boldsymbol{r}_i, \boldsymbol{m}_i, \boldsymbol{t}_i)\}}{\sum_j \exp\{\gamma \, s(\boldsymbol{r}_i, \boldsymbol{m}_i, \boldsymbol{t}_j)\}}, \tag{3}$$

where $\gamma$ is a learnable temperature parameter and $s(\boldsymbol{r}_i, \boldsymbol{m}_j, \boldsymbol{t}_k)$ is the "compatibility" score of the triplet $(i, j, k)$ in $[1, B]^3$. For ARTEMIS, this compatibility score is the sum of the EM and IS scores, respectively obtained by each of the modules presented above:

$$s(\boldsymbol{r}_i, \boldsymbol{m}_j, \boldsymbol{t}_k) = s_{EM}(\boldsymbol{m}_j, \boldsymbol{t}_k) + s_{IS}(\boldsymbol{r}_i, \boldsymbol{m}_j, \boldsymbol{t}_k). \tag{4}$$

The BBC loss, proposed as InfoNCE in the self-supervised learning literature (Oord et al., 2018), was introduced for image retrieval with text modifiers by Vo et al. (2019) and its efficacy for the task was confirmed by Lee et al. (2021). In contrast to the triplet and the soft triplet losses which rely on negative mining strategies to obtain optimal results, the BBC loss considers all the negative samples in a mini-batch, and therefore simultaneously learns from easy and hard negatives.

**Encoders.** Our approach is generic and can accommodate for diverse visual encoders $\phi$ and textual encoders $\theta$. We present results with several combinations, for the sake of comparison with prior work. We typically use GloVe word embeddings (Pennington et al., 2014) followed by either a LSTM (Hochreiter & Schmidhuber, 1997) or a BiGRU (Cho et al., 2014) as text encoder and an ImageNet pretrained ResNet18 or ResNet50 model (He et al., 2016) as image encoder. Further implementation details can be found in Appendix § A.

## 4 EXPERIMENTS

### 4.1 EXPERIMENTAL SETTING

**Datasets.** We consider two datasets focusing on the fashion domain and one on open-domain images, all three using human-written textual modifiers in natural language. The *Fashion IQ* dataset (Wu et al., 2021) is composed of 46.6k training images and around 15.5k images for both the validation and test sets. There are 18k training queries, and 12k queries per evaluation split, covering three fashion categories: women's tops (toptee), women's dresses (dress) and men's shirts

Table 1: **Ablative study** of the different components of our method. Overall $1^{st}/2^{nd}$ in **black/blue**.

| Flavors of ARTEMIS | FIQ val split | | | Shoes | | | CIRR val split | | |
|---|---|---|---|---|---|---|---|---|---|
| | $R@10$ | $R@50$ | m. rank | $R@10$ | $R@50$ | m. rank | $R@5$ | $R@50$ | $R_s@1$ |
| Image only $\langle r \mid t \rangle$ | 4.86 | 12.01 | 917.11 | 28.47 | 53.13 | 41.67 | 30.10 | 75.75 | 20.84 |
| Text only $\langle m \mid t \rangle$ | 15.55 | 36.11 | 121.44 | 13.50 | 31.46 | 154.00 | 21.93 | 65.71 | 38.28 |
| Late fusion $\langle r + m \mid t \rangle$ | **25.69** | **50.14** | **51.78** | **48.95** | **75.62** | **11.33** | 30.94 | **78.04** | 21.65 |
| IS module only | 6.18 | 16.42 | 449.56 | 32.63 | 57.41 | 31.33 | **32.28** | 77.50 | 21.16 |
| EM module only | 15.61 | 36.43 | 113.89 | 13.72 | 32.39 | 146.67 | 29.71 | 72.24 | **43.48** |
| Full ARTEMIS | **26.05** | **50.29** | **52.89** | **53.11** | **79.31** | **8.67** | **48.95** | **89.19** | **41.42** |

(shirt). The text modifier is composed of two relative captions produced by two different human annotators, exposed to the same reference-target image pair. The ***Shoes*** dataset (Guo et al., 2018) is extracted from the Attribute Discovery Dataset (Berg et al., 2010). It consists of 10k training images structured in 9k training triplets, and 4.7k test images including 1.7k test queries. The recently released ***CIRR*** dataset (Liu et al., 2021) is composed of 36k pairs of open-domain images, arranged in a 80%-10%-10% split between the train/validation/test. The annotation process is such that the modifying text should only be relevant to one image pair, and irrelevant to any other image pairs containing the same reference image.

**Evaluation.** We use the standard evaluation protocol for each dataset. We report Recall@K ($R@K$), which is the percentage of queries for which at least one of the correct ground truth item is ranked among the top K retrieved items. All reported numbers are obtained as an average of 3 runs. In Section 4.3, we additionally report standard deviations.

For FashionIQ, the standard evaluation metric is the one used for the companion challenge (Gao & Guo, 2020). It computes the $R@10$ and $R@50$ for each of the 3 fashion categories and averages these six results. For Shoes, we report the $R@1$, $R@10$ and $R@50$ as well as their average. For CIRR, following Liu et al. (2021), we also report $Recall_{subset}@K$, in which the set of candidate target images is restricted to images semantically similar to the correct target image. According to Liu et al. (2021), this metric is less sensitive to false-negatives.

**Cross-validation.** Since the test split annotations of FashionIQ became available only very recently, previous works only report results on the validation set, and resort to ad hoc strategies to set their hyper-parameters and early-stop criteria. In order to compare our method to previously published approaches, we provide results on both the test and validation sets by performing a bidirectional cross-validation between them: we run our experiments for a fixed number of epochs and, at the end of each epoch, we evaluate our model on both sets. We select the best performing model on the validation set to report results on the test set and vice-versa.When no validation split is available, as for Shoes, we report the results on the last checkpoint. For CIRR, we select the best checkpoint on the validation split to evaluate on their test server.

## 4.2 ABLATION STUDY

We first conduct an ablative study on the validation set of all datasets to evaluate the influence of several design choices in our architecture. For FashionIQ and Shoes, we additionally report median rank of the correct target. For CIRR, we report the $R@5$ and $R_s@1$ in addition of $R@50$, since they best capture the model capabilities according to Liu et al. (2021). Results are presented in Table 1. All experiments in this section use ResNet50 and BiGRU as visual and textual encoders, trained as described in the Appendix § A. We separate our ablation studies into two parts; first we look at several baselines which do not use any attention mechanism, then we challenge the different module choices by training them independently.

**Baselines.** Image-only and text-only baselines use only one of the queries elements to retrieve the target. Their compatibility scores can be written as $s_{img}(r, m, t) = \langle r \mid t \rangle$ and $s_{txt}(r, m, t) = \langle m \mid t \rangle$, respectively. We also report the late fusion of the two query embeddings, which can be seen as an attention-free version of ARTEMIS. In this case, the compatibility score becomes $s_{lf}(r, m, t) = \langle r + m \mid t \rangle$.

Table 2: **Fashion IQ, official validation set**. We report the challenge metric (**CM**) and individual $R@K$ scores. † means our re-implementation. ⋆ denotes the use of additional side information (*e.g.* extra captions from other datasets) at train time. Unless mentioned otherwise, each method uses Resnet50 and LSTM as visual and textual backbones, respectively. Overall $1^{st}/2^{nd}$ in **black**/**blue**.

| Method | CM | R@10 | | | | R@50 | | | |
|---|---|---|---|---|---|---|---|---|---|
| | | Dress | Shirt | Toptee | Mean | Dress | Shirt | Toptee | Mean |
| JVSM⋆ (Chen & Bazzani, 2020) | 19.27 | 10.70 | 12.00 | 13.00 | 11.90 | 25.90 | 27.10 | 26.90 | 26.63 |
| ComposeAE (Anwaar et al., 2021) | 20.60 | - | - | - | 11.80 | - | - | - | 29.40 |
| TCIR (Chawla et al., 2021) | 29.51 | 19.33 | 14.47 | 19.73 | 17.84 | 43.52 | 35.47 | 44.56 | 41.18 |
| CIRPLANT (Liu et al., 2021) | 25.17 | 14.38 | 13.64 | 16.44 | 14.82 | 34.66 | 33.56 | 38.34 | 35.52 |
| CIRPLANT⋆ (Liu et al., 2021) | 30.20 | 17.45 | 17.53 | 21.64 | 18.87 | 40.41 | 38.81 | 45.38 | 41.53 |
| TIRG† (RN50 + LSTM) | $36.16_{\pm0.22}$ | $24.19_{\pm0.04}$ | $20.13_{\pm0.54}$ | $26.05_{\pm0.95}$ | $23.46_{\pm0.31}$ | $50.42_{\pm1.19}$ | $43.56_{\pm0.47}$ | $52.57_{\pm0.33}$ | $48.85_{\pm0.27}$ |
| TIRG† (RN50 + BiGRU) | $35.32_{\pm0.74}$ | $23.80_{\pm1.55}$ | $19.90_{\pm0.62}$ | $25.82_{\pm0.73}$ | $23.17_{\pm0.70}$ | $48.64_{\pm1.22}$ | $42.14_{\pm1.65}$ | $51.64_{\pm0.30}$ | $47.48_{\pm0.83}$ |
| VAL (Chen et al., 2020) | 33.82 | 21.12 | 21.03 | 25.64 | 22.60 | 42.19 | 43.44 | 49.49 | 45.04 |
| VAL⋆ (Chen et al., 2020) | 35.38 | 22.53 | **22.38** | **27.53** | 24.15 | 44.00 | **44.15** | 51.68 | 46.61 |
| CoSMo (Lee et al., 2021) | 31.26 | 21.39 | 16.90 | 21.32 | 19.87 | 44.45 | 37.49 | 46.02 | 42.65 |
| ARTEMIS (RN18 + LSTM) (ours) | $34.70_{\pm0.10}$ | $25.23_{\pm0.36}$ | $20.35_{\pm0.56}$ | $23.36_{\pm0.22}$ | $22.98_{\pm0.07}$ | $48.64_{\pm0.61}$ | $43.67_{\pm0.94}$ | $46.97_{\pm0.47}$ | $46.43_{\pm0.13}$ |
| ARTEMIS (RN50 + LSTM) (ours) | $\mathbf{36.51}_{\pm0.46}$ | $\mathbf{27.34}_{\pm0.44}$ | $21.05_{\pm1.89}$ | $24.91_{\pm0.57}$ | $\mathbf{24.43}_{\pm0.61}$ | $\mathbf{51.71}_{\pm0.78}$ | $44.18_{\pm0.39}$ | $49.87_{\pm0.56}$ | $48.59_{\pm0.40}$ |
| ARTEMIS (RN18 + BiGRU) (ours) | $34.75_{\pm0.03}$ | $24.84_{\pm0.06}$ | $20.40_{\pm0.11}$ | $23.63_{\pm0.05}$ | $22.95_{\pm0.02}$ | $49.00_{\pm0.15}$ | $43.22_{\pm0.04}$ | $47.39_{\pm0.58}$ | $46.54_{\pm0.04}$ |
| ARTEMIS (RN50 + BiGRU) (ours) | $\mathbf{38.17}_{\pm0.35}$ | $27.16_{\pm0.52}$ | $21.78_{\pm0.26}$ | $29.20_{\pm0.69}$ | $26.05_{\pm0.33}$ | $52.40_{\pm0.20}$ | $43.64_{\pm1.01}$ | $54.83_{\pm0.30}$ | $50.29_{\pm0.40}$ |

Table 3: **Fashion IQ, test set**. We report the challenge metric (**CM**) and individual $R@K$ scores. † means our re-implementation. ⋆ denotes the use of additional side information (*e.g.* attributes) at training time. Overall $1^{st}/2^{nd}$ in **black**/**blue**.

| Method | CM | R@10 | | | | R@50 | | | |
|---|---|---|---|---|---|---|---|---|---|
| | | Dress | Shirt | Toptee | Mean | Dress | Shirt | Toptee | Mean |
| Transf. Dialog (Wu et al., 2021) | 21.57 | 12.45 | 11.05 | 11.24 | 11.58 | 35.21 | 28.99 | 30.45 | 31.55 |
| Transf. Dialog⋆ (Wu et al., 2021) | 22.05 | 13.39 | 11.03 | 11.74 | 12.05 | 35.56 | 29.03 | 31.52 | 32.04 |
| TIRG† (RN50 + LSTM) | $35.52_{\pm0.02}$ | $25.66_{\pm0.14}$ | $20.35_{\pm0.05}$ | $24.64_{\pm0.08}$ | $23.55_{\pm0.05}$ | $49.81_{\pm0.06}$ | $43.36_{\pm0.19}$ | $49.31_{\pm0.10}$ | $47.49_{\pm0.00}$ |
| TIRG† (RN50 + BiGRU) | $34.86_{\pm0.23}$ | $25.01_{\pm0.72}$ | $19.48_{\pm1.13}$ | $24.43_{\pm0.02}$ | $22.97_{\pm0.32}$ | $49.17_{\pm0.10}$ | $42.48_{\pm0.09}$ | $48.61_{\pm0.73}$ | $46.75_{\pm0.17}$ |
| ARTEMIS (RN18 + LSTM) (ours) | $35.05_{\pm0.11}$ | $23.57_{\pm0.08}$ | $20.09_{\pm0.15}$ | $25.26_{\pm0.16}$ | $22.97_{\pm0.04}$ | $48.15_{\pm0.04}$ | $42.67_{\pm0.14}$ | $50.57_{\pm0.86}$ | $47.13_{\pm0.24}$ |
| ARTEMIS (RN50 + LSTM) (ours) | $\mathbf{36.96}_{\pm0.34}$ | $\mathbf{26.60}_{\pm0.35}$ | $\mathbf{20.69}_{\pm0.74}$ | $27.11_{\pm0.00}$ | $\mathbf{24.80}_{\pm0.21}$ | $\mathbf{51.45}_{\pm0.77}$ | $43.13_{\pm0.80}$ | $\mathbf{52.76}_{\pm0.16}$ | $\mathbf{49.11}_{\pm0.50}$ |
| ARTEMIS (RN18 + BiGRU) (ours) | $35.36_{\pm0.08}$ | $23.92_{\pm0.06}$ | $20.19_{\pm0.13}$ | $25.80_{\pm0.20}$ | $23.30_{\pm0.04}$ | $48.71_{\pm0.18}$ | $42.01_{\pm0.28}$ | $\mathbf{51.54}_{\pm0.50}$ | $47.42_{\pm0.15}$ |
| ARTEMIS (RN50 + BiGRU) (ours) | $\mathbf{37.13}_{\pm0.03}$ | $28.13_{\pm0.36}$ | $21.43_{\pm0.22}$ | $25.91_{\pm0.02}$ | $25.16_{\pm0.01}$ | $51.66_{\pm0.24}$ | $45.22_{\pm0.02}$ | $50.41_{\pm0.86}$ | $\mathbf{49.10}_{\pm0.19}$ |

The performance of these baselines changes according to the nature of each dataset. For FashionIQ, where 68% of the modifiers make direct reference to the target image (Wu et al., 2021), the text-only baseline outperforms the image-only one. For Shoes, based on visual inspection, we conjecture that the reference-target image pairs are, on average, more similar than for FashionIQ, making the reference image more crucial for many queries. For CIRR, while querying only with the reference image offers a strong baseline, the text modifier is more discriminative when ranking on the restricted set of very similar images. This is measured by the $R_s@1$ metric. Note that the late fusion baseline produces strong results for both FashionIQ and Shoes, but barely improves upon the image-only baseline for CIRR, which highlights the importance of the attention mechanisms.

**Independent modules.** We evaluate the IS and EM independently as separate retrieval models. Their corresponding compatibility scores are given by Equations 1 and 2, respectively. We observe that IS and EM perform differently for different datasets, and as expected, follow the previous observations made for the image-only and text-only baselines. We see that both our modules slightly outperform their corresponding baseline. The full ARTEMIS performs best for all three datasets. This highlights the synergy between IS and EM, supporting the motivations extensively discussed in Sections 3.1 and 3.2, but also the fact that the method is robust to different types of datasets/ queries: those where the visual part is more important as well as those where the textual part is preponderant.

### 4.3 COMPARISON WITH THE STATE OF THE ART

We now compare ARTEMIS to state-of-the-art approaches on each benchmark. We also report results for our re-implementation of TIRG (Vo et al., 2019), a well-established composition network baseline, with the same training pipeline as ARTEMIS, as the original paper does not provide results on these more recent datasets.

Table 4: **Shoes dataset**. $\dagger$ means our re-implementation. $\star$ denotes the use of additional information (*e.g.* extra-captions, attributes) at training time. All reported methods use a ResNet50 as backbone. Overall $1^{st}/2^{nd}$ in **black**/**blue.**

| Method | $R@1$ | $R@10$ | $R@50$ | $(\sum R@K)$ /3 |
|---|---|---|---|---|
| TIRG$^{\dagger}$ (RN50-LSTM) | $15.52_{\pm 0.52}$ | $48.65_{\pm 1.03}$ | $76.49_{\pm 1.42}$ | $46.89_{\pm 0.99}$ |
| TIRG$^{\dagger}$ (RN50-BiGRU) | $14.46_{\pm 0.96}$ | $47.51_{\pm 0.68}$ | $75.17_{\pm 0.32}$ | $45.71_{\pm 0.50}$ |
| VAL (Chen et al., 2020) | 16.49 | 49.12 | 73.53 | 46.38 |
| VAL$^{\star}$ (Chen et al., 2020) | 17.18 | **51.52** | 75.83 | 48.18 |
| CoSMo (Lee et al., 2021) | 16.72 | 48.36 | 75.64 | 46.91 |
| ARTEMIS (RN50-LSTM) (ours) | **17.60**$_{\pm 0.57}$ | $51.05_{\pm 0.21}$ | **76.85**$_{\pm 0.31}$ | **48.50**$_{\pm 0.25}$ |
| ARTEMIS (RN50-BiGRU) (ours) | **18.72**$_{\pm 0.23}$ | **53.11**$_{\pm 0.77}$ | **79.31**$_{\pm 0.19}$ | **50.38**$_{\pm 0.38}$ |

Table 5: **CIRR dataset, test set**. $Recall@K$ and $Recall_{subset}@K$ (according to Liu et al. (2021), $Recall_{subset}@1$ best assess fine-grained reasoning ability). Gray background is not directly comparable to our results, since it relies on a model pre-trained on a very large set of image-caption pairs. Overall $1^{st}/2^{nd}$ in **black**/**blue.**

| Method | $Recall@K$ | | | | $Recall_{subset}@K$ | | | $\frac{(R@5+R_{sub}@1)}{2}$ |
|---|---|---|---|---|---|---|---|---|
| | $K=1$ | $K=5$ | $K=10$ | $K=50$ | $K=1$ | $K=2$ | $K=3$ | |
| CIRPLANT$^{\star}$ (init. OSCAR) | 19.55 | 52.55 | 68.39 | 92.38 | 39.20 | 63.03 | 79.49 | 45.88 |
| CIRPLANT | **15.18** | **43.36** | **60.48** | **87.64** | 33.81 | 56.99 | **75.40** | **38.59** |
| TIRG$^{\dagger}$ (BiGRU) | $10.01_{\pm 0.34}$ | $38.31_{\pm 0.55}$ | $54.59_{\pm 0.59}$ | $84.69_{\pm 0.78}$ | **37.36**$_{\pm 0.06}$ | **59.31**$_{\pm 0.53}$ | $72.51_{\pm 0.80}$ | $37.84_{\pm 0.27}$ |
| ARTEMIS (BIGRU) (ours) | **16.96**$_{\pm 0.43}$ | **46.10**$_{\pm 0.36}$ | **61.31**$_{\pm 1.30}$ | **87.73**$_{\pm 0.62}$ | **39.99**$_{\pm 0.65}$ | **62.20**$_{\pm 0.78}$ | **75.67**$_{\pm 0.75}$ | **43.05**$_{\pm 0.51}$ |

**Results on Fashion IQ** are reported in Table 2 and Table 3 for the validation and the test sets, respectively. Since the test split annotations were unavailable until very recently (only Wu et al. (2021) published results on that split), most published approaches report results on the validation set. In both cases our best ARTEMIS (RN50 + BiGRU) outperforms the state of the art.

As most published works (Chen et al., 2020; Lee et al., 2021) use LSTM as text backbone and ResNet50 as image backbone, we also report results for ARTEMIS in this setting and show that we remain competitive, outperforming VAL$^{\star}$ by an average of 1.1% over the challenge metric. Furthermore, we also report results with ResNet18 to compare to several recent works (Anwaar et al., 2021; Chen & Bazzani, 2020), who prioritize a faster and smaller backbone. We observe that, for smaller image backbone, LSTM seems to perform better then BiGRU. This is also true for our re-implementation of TIRG.

**Results on Shoes** are presented in Table 4. As the Shoes dataset does not contain a dedicated validation set, we reuse the exact same training procedure and hyperparameters as for FashionIQ. ARTEMIS obtains state-of-the-art results with both LSTM and BiGRU encoders, confirming our approach's ability to generalize.

**CIRR results** are reported in Table 5. As recommended by Liu et al. (2021), we use pre-extracted ResNet152 features for the images. This affects our training pipeline as the image backbone cannot be finetuned and we cannot apply any image transformation. CIRPLANT obtains competitive results only when its multi-layer cross-modal transformer is initialized with the weights from the Oscar model (Li et al., 2020) trained on 6.5 million image-caption pairs. When compared to methods trained on the same data, ARTEMIS outperforms CIRPLANT on all metrics, in spite of a simpler architecture. In particular, our $R_s@1$ result suggests, according to Liu et al. (2021), that ARTEMIS has a great fine-grained reasoning ability.

## 4.4 QUALITATIVE RESULTS

In Figure 3 we present a few heatmap examples obtained with the Grad-CAM (Selvaraju et al., 2017) algorithm. Specifically, the image activations at the last convolutional layer of the CNN are reweighed with the pooled gradients of the same layer, after back-propagation of the coefficients contributing the most to the EM and IS scores. It appears clearly that the IS module attends to the

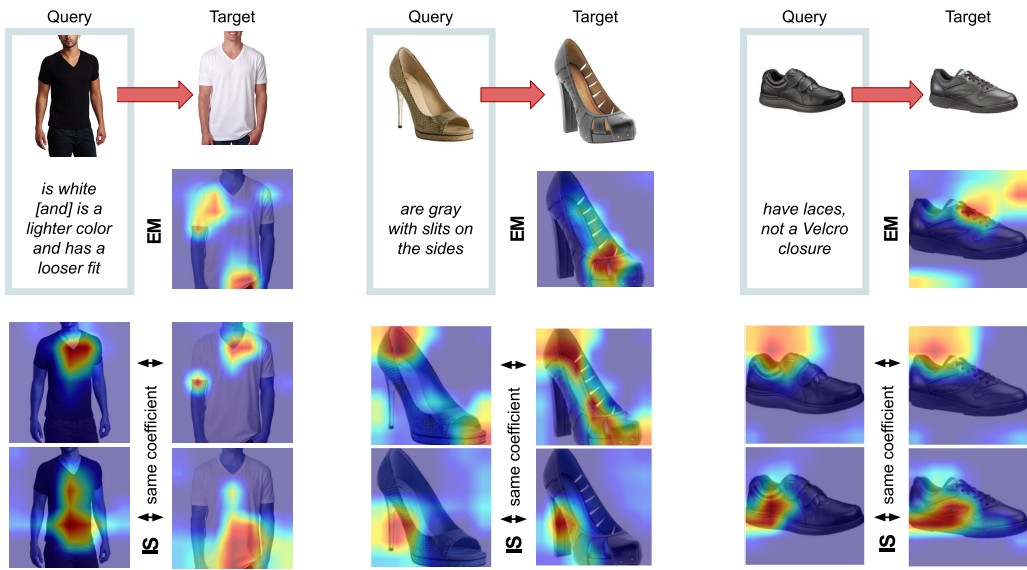

Figure 3: Visualisation of image parts contributing the most to the EM and IS scores, for queries from the Fashion IQ and Shoes datasets where the correct target is ranked first. An EM component is presented on the target image. The bottom two rows show two relevant components for IS, applied to both the source and the target images (see Appendix § D for details).

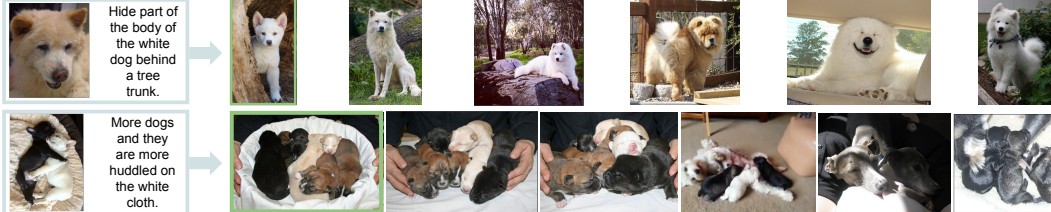

Figure 4: Qualitative examples of image-text queries of CIRR and its Top-6 retrieved results. A green frame denotes the ground-truth target (see Appendix § D for more examples).

visual cues that are shared between the reference and the target images, such as the shirt's collar and bottom, the sole shape & the heel, or the color and the shoe opening. In a complementary way, the EM module addresses the image parts that are the most related to the caption: the color, the shirt's fit, the slits on the side or the shoe's closure.

In Figure 4, we present some query examples from the CIRR dataset along with the Top-6 retrieved images. In both examples, we see that the reference image contributes a strong semantic context, and that the

text modifier provides important information to distinguish the correct target image (framed in green) from other very similar images. Additional qualitative examples and further analysis are available in the Appendix § D.

## 5 CONCLUSION

We proposed ARTEMIS, a new method for image search with free-form text modifiers. It combines two modules, each focusing on one of the modalities of the query. The *Explicit Matching* module assesses how potential targets fit the textual modifier while the *Implicit Similarity* module compares potential target images to the reference image, assisted by the text. Both modules contain a light-weight attention mechanism which uses the text to guide the retrieval process in a way that is specific to each modality of the query. Extensive experiments on several benchmarks validate our approach and show that we consistently reach state-of-the-art performance.

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

APPENDIX

This document provides additional results that complement the main paper. Its content includes some implementation details (§ A), our experiments on an additional dataset (§ B), an efficiency study of our method in comparison to other approaches (§ C), and some extra qualitative results for ARTEMIS (§ D).

## A    IMPLEMENTATION DETAILS

Texts are pre-processed to replace special characters by spaces and to remove all other characters than letters. We observe that using spelling correction and lemmatization does not improve the results. The text modifier is first tokenized into words, and encoded using 300-dimensional GloVe embeddings (Pennington et al., 2014). Either a BiGRU (Cho et al., 2014), or a LSTM (Hochreiter & Schmidhuber, 1997) (followed by an average pooling and a fully-connected layer, as in Chen et al. (2020)) are learned from scratch on top of the word embeddings to produce a sentence-level feature. For Fashion IQ (Wu et al., 2021), the only dataset for which each reference/target image pair is provided with two relative captions (used as text modifiers) from two different annotators, we follow previous works (Lee et al., 2021) and use the concatenation of the two captions (in both orders) for training and evaluation. This means that we use about 12k queries fr the validation split.

To maintain the aspect ratio and avoid losing information about the top and the bottom of the clothing items, images are first padded with white pixels to obtain a square. They are then resized to $256 \times 256$ pixels and augmented using random horizontal flip and a random crop of size of $224 \times 224$ pixels during training. For inference, images are resized to $256 \times 256$ and center cropped to $224 \times 224$. The image features are obtained using a CNN with a GeM pooling (Radenović et al., 2018), which is widely used in the instance-level image retrieval community instead of the usual average pooling. The pooling layer is followed by a newly learned fully-connected layer. Our experiments employ either a ResNet18 or ResNet50 architecture (He et al., 2016), initialized on ImageNet (Russakovsky et al., 2015). The only exceptions are the experiments on CIRR, for which not all images are available to download. Liu et al. (2021) provide pre-extracted ResNet152 features (outputs of the average pooling layer) that we use as a replacement of the image backbone. As a consequence, for all experiments on CIRR, the image backbone is not fine-tuned and we do not apply any image transformation.

Following Song & Soleymani (2019), we freeze the base encoders during the first 8 epochs to pre-train the sentence encoder, as well as the EM and IS modules. Then, we train our model end-to-end for 50 epochs. Our training pipeline uses AdamW optimizer (Loshchilov & Hutter, 2017), a batch size of 32 and an initial learning rate of $5 \times 10^{-4}$ with a decay of $0.5$ every 10 epochs. The dimension of both the image and the textual embeddings is set to $H_T = H_I = 512$.

## B    EXPERIMENTS ON FASHION200K

In the literature (Vo et al., 2019; Chen et al., 2020; Lee et al., 2021), the Fashion200K dataset (Han et al., 2017) is used as a dataset for image search with text modifiers. The dataset is composed of 201k in-shop fashion article images, split into 172k train images and 29k test images. From these test images, Vo et al. (2019) provides a list of 31k image pairing which is used to produce candidate queries for evaluation. The text modifiers, in contrast to the other presented datasets, are not in natural language: they are automatically generated from the set of image attributes following the "*replace att. X by att. Y*" template. In our work, we focus on free-form human annotations, which is the common point between the three datasets presented in the main paper. However, we here compare ARTEMIS to the other published works on this dataset, for reference.

In order to better handle the large train set of Fashion200K, we follow VAL (Chen et al., 2020) and CoSMo (Lee et al., 2021), and we train our model with a larger mini-batch of 128 for 100 epochs end-to-end. While Chen et al. (2020); Chen & Bazzani (2020) use MobileNet-v1 as image backbone, we follow Vo et al. (2019); Lee et al. (2021) and use ResNet18. As Lee et al. (2021) pointed out, though the difference in model architecture may play a role in overall results, both models have almost identical scores on both ImageNet Top1 and Top5 error rates.

Table 6: **Fashion200K dataset results**. [†] means results of our re-implementation. [‡] denotes results published by Vo et al. (2019). [⋆] denotes the use of additional information (*e.g.* extra captions) at training time. For our results, we report the average of 3 runs. Overall $1^{st}$/$2^{nd}$/$3^{rd}$ in **black/blue/red**.

| Method | $R@1$ | $R@10$ | $R@50$ | $(\sum R@K)$ /3 |
|---|---|---|---|---|
| FiLM[‡] (Perez et al., 2018) | 12.9 | 39.5 | 61.9 | 38.1 |
| MRN[‡] (Kim et al., 2016) | 13.4 | 40.0 | 61.9 | 38.4 |
| Relationship[‡] (Santoro et al., 2017) | 13.0 | 40.5 | 62.4 | 38.6 |
| TIRG[‡] (Vo et al., 2019) | 14.1 | 42.5 | 63.8 | 40.1 |
| TIRG[†] (RN18+LSTM) | $20.8_{\pm2.5}$ | $50.7_{\pm0.4}$ | $69.9_{\pm0.8}$ | $47.1_{\pm1.0}$ |
| TIRG[†] (RN18+BiGRU) | $20.2_{\pm0.4}$ | $49.4_{\pm1.2}$ | $69.8_{\pm1.2}$ | $46.5_{\pm0.6}$ |
| LBF (Hosseinzadeh & Wang, 2020) | 17.8 | 48.4 | 68.5 | 44.9 |
| VAL (Chen et al., 2020) | 21.2 | 49.0 | 68.8 | 46.3 |
| VAL[⋆] (Chen et al., 2020) | **22.9** | **50.8** | **72.7** | **48.8** |
| CoSMo (Lee et al., 2021) | **23.3** | 50.4 | 69.3 | **47.7** |
| JVSM[⋆] (Chen & Bazzani, 2020) | 19.0 | **52.1** | **70.0** | 47.0 |
| ARTEMIS (ours) (RN18+LSTM) | **$21.5_{\pm0.9}$** | **$51.1_{\pm1.0}$** | **$70.5_{\pm0.8}$** | **$47.7_{\pm0.8}$** |
| ARTEMIS (ours) (RN18+BiGRU) | $20.2_{\pm0.9}$ | $49.3_{\pm0.4}$ | $69.3_{\pm0.9}$ | $46.2_{\pm0.5}$ |

Interestingly, while ARTEMIS performed better with a BiGRU as text encoder for all other datasets, our best results for Fashion200K are obtained with a LSTM. We observe the same for our implementation of TIRG, and more generally for all our experiments that use a ResNet18.

We observe on Table 6 that VAL[⋆] obtains the best performance, thanks to the extra captions used to pre-train their auxiliary regularizer. The same side information was also leveraged by Chen & Bazzani (2020). Excluding models trained with this extra data, both ARTEMIS and CoSMo outperform VAL, and our approach obtain the best results in 2 out of the 4 evaluation metrics.

## C COMPLEXITY AND EFFICIENCY STUDY

In the main paper, we extensively discuss our method's performance in terms of accuracy, on several benchmarks. This section provides a complementary study that compares ARTEMIS' complexity with the complexity of several approaches. Since no single measure is enough to assess the complexity of a model (Dehghani et al., 2021), we compare the different models on three metrics:

- **Number of parameters.** We measure the number of trainable parameters of the full models. Many works use this metric as a proxy to model complexity and memory footprint.

- **Number of multiply-add operations.** The number of multiply-accumulate operations (MAC) is a theoretical measure that allows to compare the number of operations needed by the model to perform a forward pass, independently of the hardware (Johnson, 2018). We report the number of GMACs for the forward pass of a triplet $(I_r, T_m, I_t)$, where reference and target images are both of size $224 \times 224$, and the modifier text is composed of 20 tokens (in average, CIRR sentences contain 11.3 words and FashionIQ 5.3, according to Table 1 in Liu et al. (2021)).

- **Latency time.** Latency usually refers to the required time to perform an inference forward pass, either for a batch of images or the full dataset. It is an informative speed metric for real-time systems that require user input, so it is well suited to the retrieval task. We measure the time each method needs to process all queries, all target images, and all query-target scores of the FashionIQ validation split. Note that 12k queries are involved, since we do the inference for the concatenation of the two captions in both orders: 'cap1 and cap2', 'cap2 and cap1', following Lee et al. (2021) (see details in Appendix §A). All latency times are measured on the same GPU NVIDIA T4.

We report results for ARTEMIS, TIRG, VAL and CoSMo. For a fair comparison, all reported numbers are obtained for ResNet50 and LSTM as image and text encoder respectively. Since these encoders are an important source of complexity in the models, we also evaluate the encoders alone,

Table 7: **Efficiency comparison.** All models use Resnet50 as image encoder and LSTM as text encoder. In red, we give the added values compared to our reference point which only considers the encoders. Latency is computed over the 12k queries of the FashionIQ validation set.

| Method | MACs (G) | Parameters (M) | Latency FIQ (s) |
|---|---|---|---|
| Reference (only RN50 + LSTM encoders) | 8.35 | 31.65 | 114.09 |
| ARTEMIS | 8.35 (+0.001) | 32.96 (+1.31) | 122.87 (+8.78) |
| TIRG (Vo et al., 2019) | 8.35 (+0.002) | 33.75 (+2.10) | 118.20 (+4.11) |
| VAL (Chen et al., 2020) | 8.39 (+0.040) | 59.48 (+27.83) | - |
| CoSMo (Lee et al., 2021) | 10.45 (+2.110) | 79.32 (+47.73) | 133.84 (+19.75) |

to provide a reference point. For this reference, we estimate the corresponding latency by evaluating the late-fusion baseline described in Section 4.2. We report our results in Table 7. We make the following observations:

**ARTEMIS and TIRG both add very little complexity compared to the reference.** Similarly to TIRG (Vo et al., 2019), our approach adds only a few operations and parameters to the base encoders used to compute $r$, $m$ and $t$. Both methods' parameters mostly come from their fully-connected layers. TIRG uses more parameters than ARTEMIS because its gating mechanisms takes the concatenated vector $[r, m]$ as input instead of just the text embeddings $m$, doubling the number of required connections with respect to ARTEMIS .

**ARTEMIS and TIRG add some reasonable extra-latency compared to the reference.** Both ARTEMIS and TIRG add little latency time to the encoder reference point. TIRG is more efficient due to its simple pass on the target representation: pre-computed $t$ can be directly compared to the composite query representation by a simple cosine similarity. ARTEMIS also leverages pre-computed $t$, but re-weights its dimensions using $\mathcal{A}_{IS}(m)$ and $\mathcal{A}_{EM}(m)$ then L2-normalizes the result before comparing to the query features, with cosine similarity as well. This double use of the target explains ARTEMIS ' slightly longer latency time.

**VAL has a higher computational complexity.** We were unable to produce the same experiments on the publicly available code of VAL to compute the latency. For the complexity measurements, we can attest to an increase in the number of GMACs and parameters that is more significant than for both ARTEMIS and TIRG. The latency is likely to increase similarly, as VAL inserted composite transformers at different levels of the visual encoder, and compares representations coming from the concatenation of average-pooled features at low-, mid- and high-level (*i.e.* features of size $2048 + 1024 + 512$).

**CoSMo adds both extra-complexity and extra-latency.** CoSMo, similarly to TIRG, uses a composition module to fuse $r$ and $m$, and does a single pass on each target. However, its composition module, combining a disentangled non-local block for content modulation and text-guided affine transformations for style modulation, is more complex than ARTEMIS . Furthermore, while ARTEMIS and TIRG introduce little complexity, CoSMo adds around 25% multiply-accumulation operations, and more than doubles the number of trainable parameters.

## D  QUALITATIVE RESULTS

**Additional ranking results.** We provide qualitative results for the FashionIQ (Wu et al., 2021), the Shoes (Guo et al., 2018; Berg et al., 2010) and the CIRR (Liu et al., 2021) datasets in Figures 5, 6, 7, 8 and 9. They illustrate the model's ability to properly encode and reason on general concepts. In the case of FashionIQ, it includes the length of sleeves and dresses, the patterns (floral, stripes, dots, plaid) or the logo color (different from the background color) and the looseness of clothing items. As for Shoes it captures shininess, wool/suede material, and the presence of laces, of fur, of Velcro closures, or of a buckle, for instance. The CIRR dataset is open-domain, and we notice that the model is able to reason on a wide variety of semantic concepts (from trees to the people's youth). The model is also able to deal with more complex information, such as dolphins (Figure 6, second row) or different renderings of the same logo (Figure 6, third row). Eventually, we observe that the

model gathers pieces of information from the reference image when needed (style and cut, length or size, collar form, sleeve length, pattern or logo, sole shape, animal breed, sculpture etc.). While the model relies on the text modifier to propose coherent images, it also resorts to the reference image to fill in the gaps.

**Limitations.** Extensive studies of the qualitative results (Figure 10) reveal that the model has trouble to understand the negative formulations: in the third row of Figure 10, the proposed toptees remain colorful; in the second row, the ranked clothing items are still loose and sheer; the last row shows a mix of shoes with laces and other with straps. For FashionIQ, it is likely due to the fact that only 3.5% of the queries use negation, according to a semantic analysis carried out in the companion paper (Wu et al., 2021). Supposedly, there are not enough examples at training time for the model to learn the associated semantic.

Besides, the captions were written by different human annotators. They sometimes present a subjective judgment (*e.g.* "is cuter", "more manly") or lack essential details to discriminate better the target among other images in the gallery. We give a few illustrative examples of such cases in Figure 10. For the top query, the specific required color is not mentioned, hence the various guesses taken by the model.

One limitation inherent to the task is well illustrated by the blue dress example in Figure 5, the boot example in Figure 8 or the bird example in Figure 9: very often, many candidate target images would be acceptable answers to a given query, beyond the one annotated as ground-truth. Hence we believe $R@K$ is a more reliable metric when K is bigger as it's more lenient to plausible wrong predictions. Other works (Musgrave et al., 2020; Chun et al., 2021) have pointed out the limitations of relying on $R@K$ for retrieval tasks when $K$ is too small, and suggested other metrics. This is also a limitation that Liu et al. (2021) tries to address with their $R_s@K$ metric, by annotating images while aware of possible false negatives.

**Additional heatmaps** on examples extracted from the FashionIQ dataset (Wu et al., 2021) are presented in Figure 11. Similarly, Figure 12 presents results on the Shoes (Guo et al., 2018; Berg et al., 2010) dataset.

The most important parts of the image with regard to the computation of each score, Explicit Matching (EM) and Implicit Similarity (IS), are highlighted in red. The coefficients of EM and IS intermediary results (*i.e.* before completing the scalar product with the sum) are referred as "components". The heatmaps are obtained by backpropagating to the last convolutional layer of the CNN some of the components that contribute the most to the EM and IS scores. We show the influence of a same IS component both on the reference and the target images, since both images are involved in the computation of the IS score. Conversely, EM is based only on the target image, with regard to the text modifier.

It emerges from the IS visualizations that the model is able to draw the parallel between the two images on several concepts. For clothing items, it can be the shape of the collar, the sleeve length or the cut of the dress. For shoe items, it includes the presence of heels or laces, the shoe opening or the sole shape. In a complementary way, the EM module addresses the image parts that are the most related to the caption: for example, it focuses on the pattern, the color or decoration requirements.

Eventually, these visualizations help to understand what semantic parts of the image are taken into account by the model to evaluate the relevance of a candidate target image for a given query.

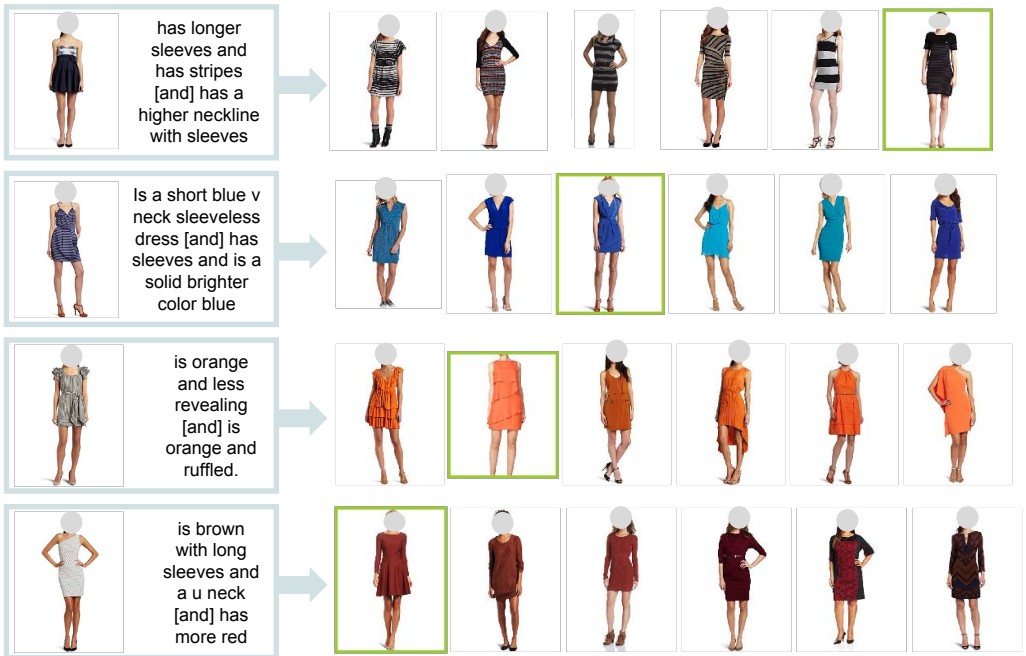

Figure 5: Qualitative results for queries from the "dress" category of the FashionIQ dataset (Wu et al., 2021). We show the six top ranked images for each query. The expected targets are indicated with a green border. It appears that the model is able to reason on several concepts (color, pattern, ruffles, sleeve length, form of the neck...), and to use the reference image to infer unspecified properties in the text modifier (style, length, tightness...).

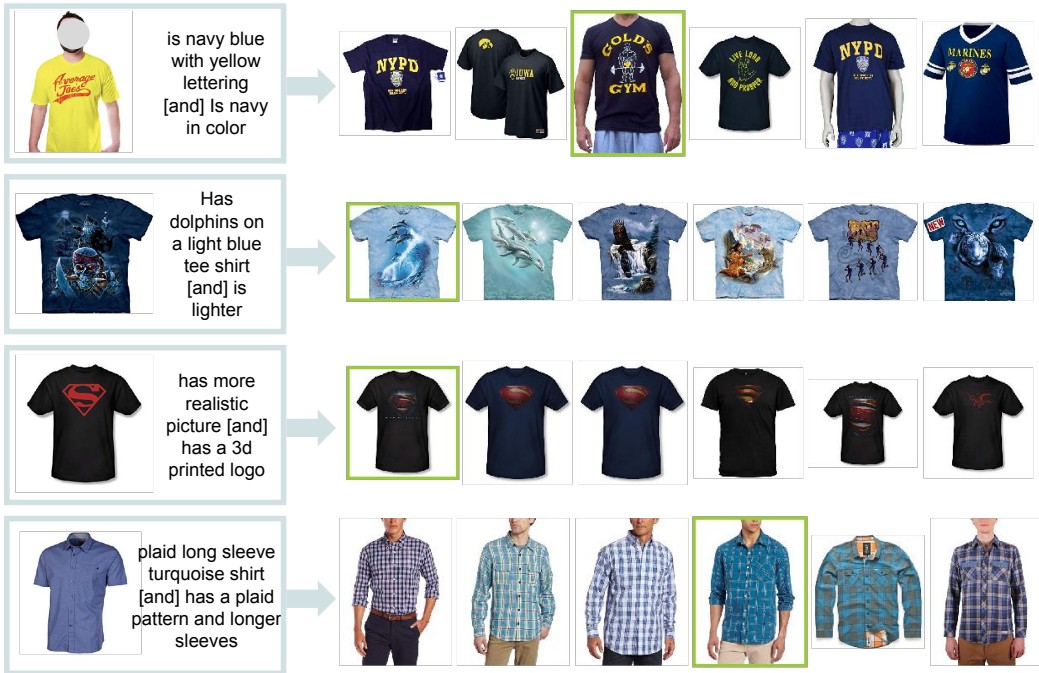

Figure 6: Qualitative results for queries from the "shirt" category of the FashionIQ dataset (Wu et al., 2021). We show the six top ranked images for each query. The expected targets are indicated with a green border. It appears that the model is able to reason on several concepts (different color for the clothing background and the logo, graphic, style, sleeve length...), and to use the reference image to infer unspecified properties in the text modifier (kind, style, form of the collar...).

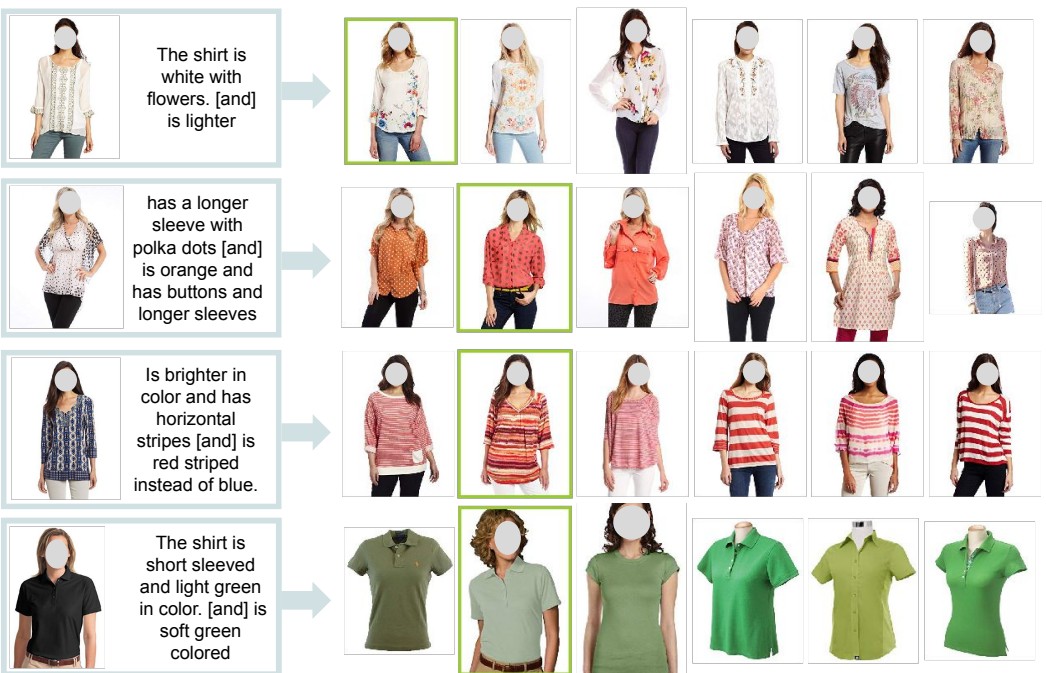

Figure 7: Qualitative results for queries from the "toptee" category of the FashionIQ dataset (Wu et al., 2021). We show the six top ranked images for each query. The expected targets are indicated with a green border. It appears that the model is able to reason on several concepts (color, pattern, presence of buttons, sleeve length, form of the neck...), and to use the reference image to infer unspecified properties in the text modifier (kind, style, length, form of the collar...).

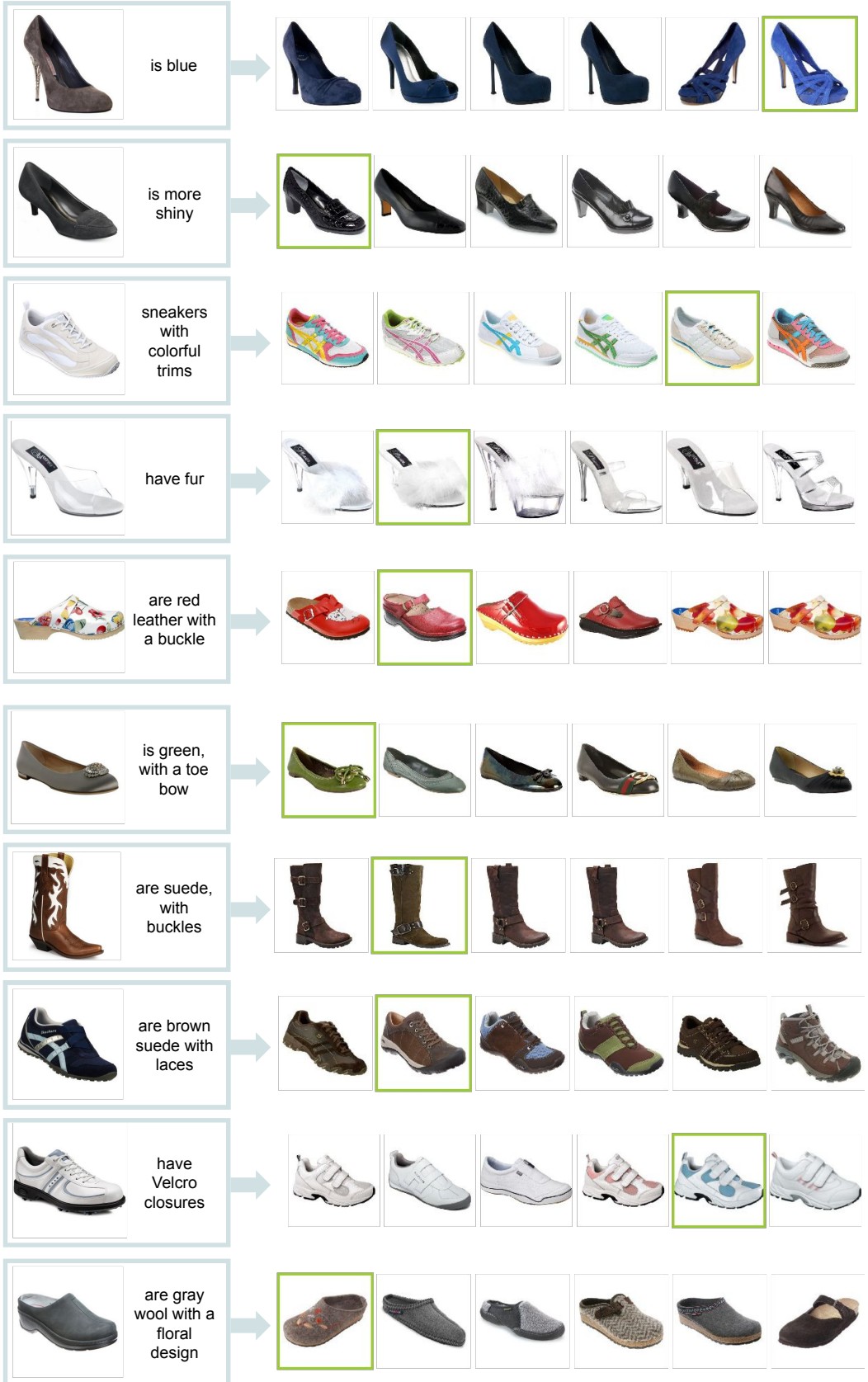

Figure 8: Qualitative results for queries from the Shoes dataset (Guo et al., 2018; Berg et al., 2010). It appears that the model is able to reason on several concepts (kind, color, style, shininess, fur, buckle, laces, Velcro closures...), and to use the reference image to infer unspecified properties in the text modifier (kind, style, presence of heels or laces, shapes of the sole and of the opening, etc.).

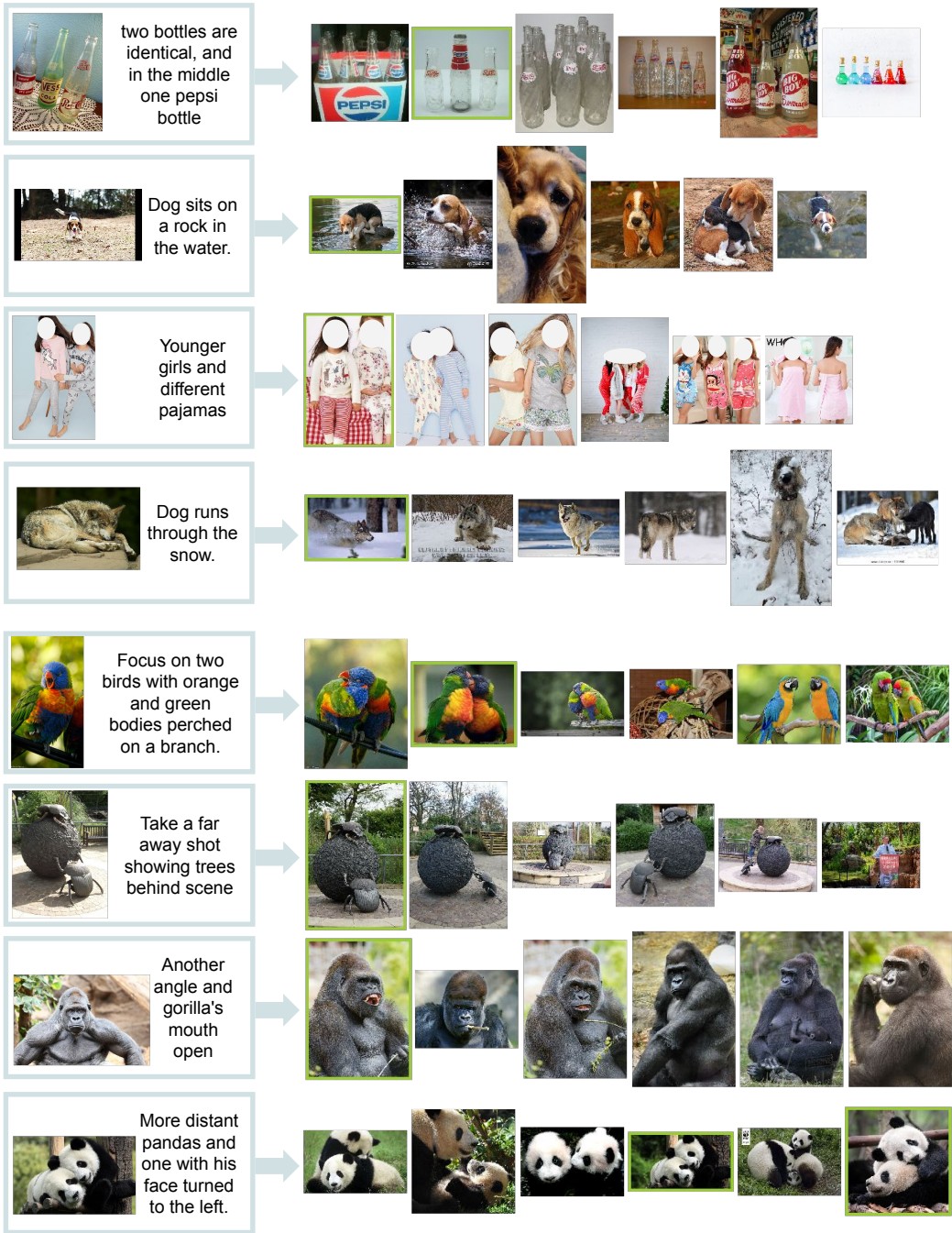

Figure 9: Qualitative results for queries from the CIRR dataset (Liu et al., 2021). We show the six top ranked images for each query. The expected targets are indicated with a green border. It appears that the model is able to reason on several concepts (plurality, background/foreground, natural elements, viewpoint...), and to use the reference image to infer unspecified properties in the text modifier (*e.g.* the animal breed).

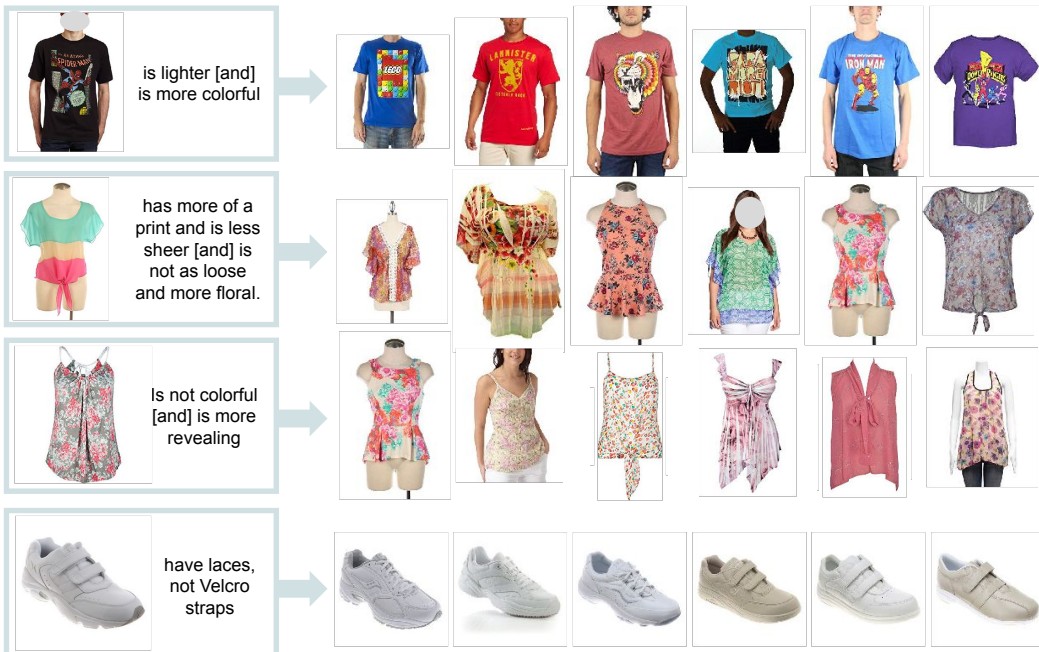

Figure 10: Qualitative results showing some limitations, for queries from FashionIQ (Wu et al., 2021) (the first 3 rows) and Shoes (Guo et al., 2018; Berg et al., 2010) (last row). For any of these examples, the ground truth is not included by the model in the top ranking of potential targets. This is either due to a lack of information (first row) or the non-understanding of negation (last 3 rows).

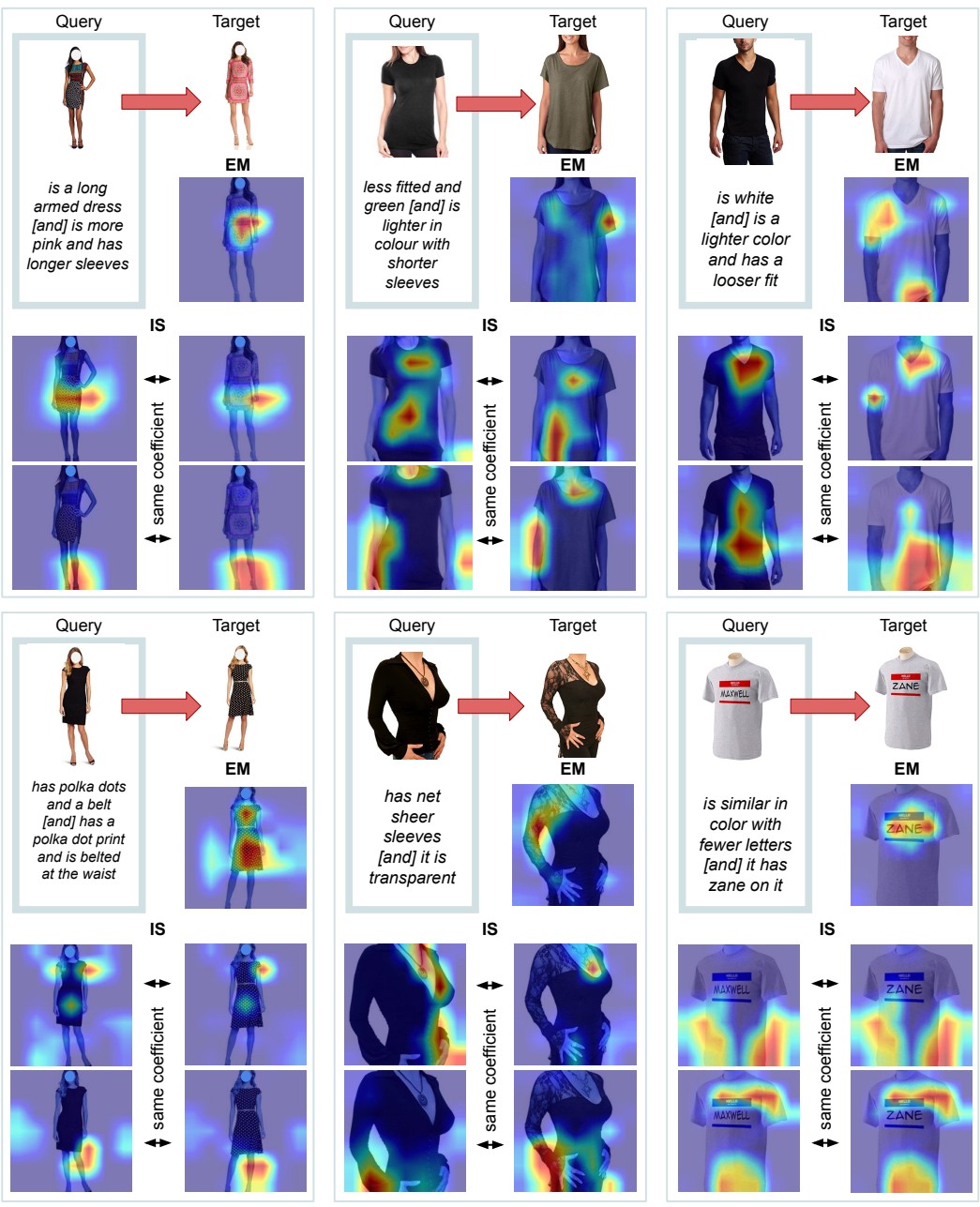

Figure 11: Visualisation of image parts contributing the most to the EM and IS scores, for queries from the FashionIQ dataset (Wu et al., 2021) where the correct target is ranked first. For each of the six blocks, we provide the heatmaps of an EM component on the target image, and of two relevant IS components, applied to both the reference and the target images. The EM heatmaps highlight some of the connections made by the model between the text modifier and the target image. The IS heatmaps display some detected similarities between the two images. Both EM and IS help to evaluate the relevance of the target image with regard to the query.

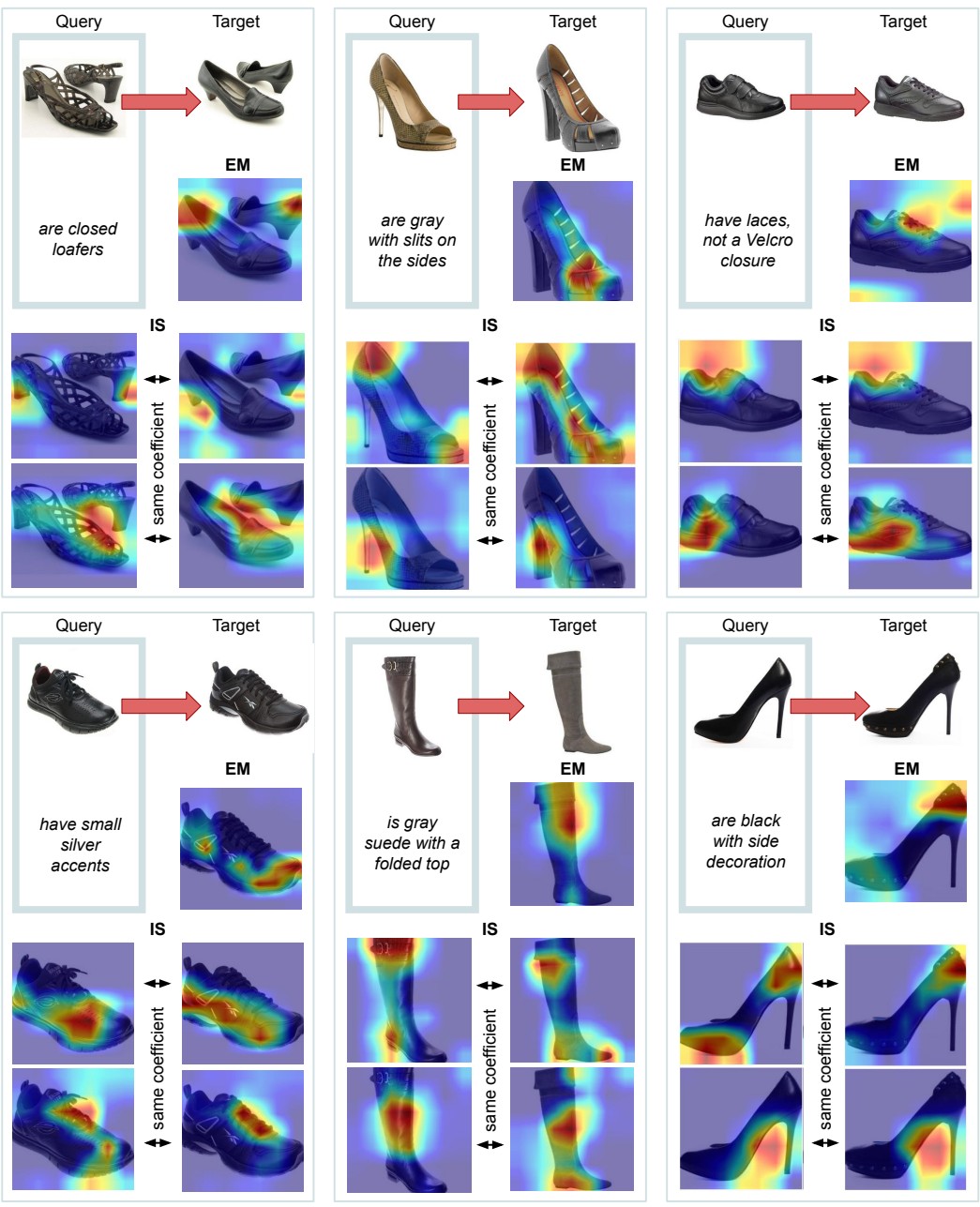

Figure 12: Visualisation of image parts contributing the most to the EM and IS scores, for queries from the Shoes dataset (Guo et al., 2018; Berg et al., 2010) where the correct target is ranked first. For each of the six blocks, we provide the heatmaps of an EM component on the target image, and of two relevant IS components, applied to both the reference and the target images. The EM heatmaps highlight some of the connections made by the model between the text modifier and the target image. The IS heatmaps display some detected similarities between the two images. Both EM and IS help to evaluate the relevance of the target image with regard to the query.

