# OpenReview forum: "ARTEMIS: Attention-based Retrieval with Text-Explicit Matching and Implicit Similarity"
_ICLR.cc/2022/Conference — ICLR 2022 Poster_

### Official Review · Reviewer_pgma · 2021-10-31

**Correctness:** 3
**Technical Novelty And Significance:** 2
**Empirical Novelty And Significance:** Not applicable
**Recommendation:** 6
**Confidence:** 4

**Main Review:**

- Strengths:

  1. It is somewhat interesting to consider the image retrieval with text modifiers in a unified framework, and the strategy of directly combining image-image similarity and image-text similarity seems easy but effective.

  2. This paper is well presented, with clear organization and detailed visual demonstration in experiments.

- Weaknesses:

  1. The attention mechanism has been widely utilized for image-text matching/VQA such as [1,2, 3], and the proposed element-wise product in IS and EM may not provide many insights.
  [1] Person Search with Natural Language Description, CVPR 2017.
  [2] Hierarchical Question-Image Co-Attention for Visual Question Answering, NIPS 2016.
  [3] Deep Attention Neural Tensor Network for Visual Question Answering, ECCV 2018.

  2. The explanations on the proposed IS and EM module are not clear enough. It would be better to provide more detailed explanations on the insights/function/effect of the specifically designed architecture.

  3. Another concern might be the framework itself. Different from CoSMo which learns feature embeddings for fast retrieval, this paper requires to inference each query-target-text triplet online, which would be much slower. It would be better to give more comparisons on efficiency and performance with different frameworks.



**Summary Of The Paper:**

This paper aims to tackle the problem of image search with free-form text modifiers and proposes to combine the implicit similarity and explicit matching score. The implicit similarity module utilizes the textual modifier as a filter to compute the similarity between the query and the target image, and the explicit matching module computes the similarity between the text and the filtered target image. This work combines the image-text similarity and text-guided image-image similarity in a unified framework, and experiments on the Fashion IQ, Shoes and CIRR dataset demonstrates the effectiveness of the proposed strategy.

**Summary Of The Review:**

Overall, this paper proposes the ARTEMIS for image-search with free-form text modifiers. Although simple and effective, the methodology may not be novel enough: it is more like a simple combination of matching scores and does not provide many insights.

---

> ### Author Response · Authors · 2021-11-17
> **Response to Rpgma**
>
> We would like to thank the reviewer for a constructive review. We are happy the reviewer found the paper “well presented, with clear organization and detailed visual demonstration in experiments”.
>
> > The attention mechanism has been widely utilized for image-text matching/VQA [1,2,3] and proposed element-wise products in IS and EM may not provide many insights.
>
> Thank you for mentioning those references where attention mechanisms have been used in methods combining text and image. We have added them in a dedicated paragraph about attention at the end of the related work section (Section 2). Note, however, that the goal of those papers and the role of their attention modules differ from those of our approach.
>
> - [1] targets cross-modal person retrieval. An attention vector is learned for each word in the query, and those are combined to build word-image affinity scores. Our approach weights the reference image or the text modifier to compare to the target image instead. Furthermore, in their case, the attention mechanism takes as input the hidden state of a LSTM applied to both the word and the image, hence the input of their attention mechanism is already cross-modal, while ours only takes the query textual modifier and is independent from the target image.
>
> - [2] addresses VQA and proposes two complex visual and textual co-attention mechanisms between the image and the question to modulate each other’s representations.  The first one computes cross-attention  between image and question features at all pairs of image-locations and question-locations, the second alternates  between generating image attention based on the question summary and question attention  based on image features. In contrast,  we learn text-guided light-weight attentions defined by a simple fully-connected layer on top of the global query-text representation.
>
> -  [3] addresses VQA and uses complex attention mechanisms. It shares relations to other papers we already discussed in Section 2 (Lee et al., 2018; Li et al., 2019; 2020, Miech et al. 2021).
>
> > Different from CoSMo which learns feature embeddings for fast retrieval, this paper requires to inference each query-target-text triplet online [..] give more comparisons on efficiency and performance
>
> Thanks for the suggestion. We agree that a quantitative evaluation of the efficiency of the different methods would strongly enrich the paper. We have added an entire section about complexity in the appendix (see section C). Please find here a subset of Table 7, comparing the complexity of ARTEMIS with CoSMo’s.
>
>
> |                        | Complexity | #Parameters | Latency |
> |------------------------|------------|-------------|---------|
> | Encoder baseline (RN50 + LSTM) | 8.35 GMac  | 31.65 M     |    71.31 s    |
> | ARTEMIS        | 8.35 GMac  | 32.96 M     |   79.52 s     |
> | CoSMo           | 10.45 GMac | 79.32 M     |  111.53 s  |
>
> From this table we can see that ARTEMIS has only a few additional parameters and almost no extra complexity compared to the encoder baseline. It also has a lower latency than CoSMo at inference time (79 seconds versus 111 seconds to obtain all query-target similarities on the FashionIQ validation set). This shows that ARTEMIS is relatively efficient. In particular, it is important to note that: (1) as for CoSMo, the visual representations from all gallery images can be pre-computed offline, and (2) the target is not used as input to any layer ; it is simply subject to a few additional pointwise products and sums.
> At inference, our approach requires processing a few additional pointwise products, whose cost is negligible for the database sizes we experimented with, but which could have an impact for truly large scale datasets. However, as it is very standard in information retrieval, such large databases would be first processed with a filtering step, selecting a first manageable set of relevant results, and only these selected results would be processed by an approach for the task at hand, such as ARTEMIS, CoSMo or any other.

---

> > ### Author Response · Authors · 2021-11-17
> > **Response to Rpgma (part 2)**
> >
> > > The explanations on the proposed IS and EM module are not clear enough [..] more detailed explanations on the insights/function/effect of the specifically designed architecture
> >
> > Thanks for this feedback. In section 3, we extensively discuss the rationale behind the design of our model architecture (see specifically the paragraphs “Two complementary views of the task.” and “Proposed approach.”). Our ablative study (section 4) and qualitative results (Figure 3) try to provide quantitative and qualitative insights about the roles of those EM & IS scores. In particular, the heatmaps (Figure 3) show the effect of EM & IS on the query and target images (“selection” of visual cues). We try to reformulate and clarify the motivation behind our components below.
> > In ARTEMIS, text plays an important role, both 1) as an element of the query in its own right, and 2) as a way to guide the retrieval process, emphasizing/weighting which part(s) of the target image each element of the query (visual and textual) should focus on. This text-driven weighting process is instantiated with our two attention vectors $\mathcal A _{EM}(m)$) and $\mathcal A _{IS}(m)$ which respectively reweight the visual representation of the query image, and the textual representation of the text modifier, leading to EM and IS scores.
> > We hope this paragraph and the pointers to our ablation and qualitative answers the reviewer concerns. If it does not, we apologize and we would be grateful for any further suggestion on the type of insights the reviewer is looking for.

---

> > > ### Comment · Reviewer_pgma · 2021-11-24
> > > **Reply to author response**
> > >
> > > I would like to thank authors for detailed response and updated manuscript. Most of my concerns have been addressed, and I will raise my vote to "weak accept".

---

### Official Review · Reviewer_L8FA · 2021-11-02

**Correctness:** 4
**Technical Novelty And Significance:** 3
**Empirical Novelty And Significance:** 4
**Recommendation:** 8
**Confidence:** 4

**Main Review:**

Strengths
-------

* Extremely well written and well motivated
* A simple but novel idea/architecture
* Comprehensive experiments showing the proposed approach achieves state-of-the-art performance on 3 datasets
* The ablations clearly show the utility of each of the parts of the already simple architecture

Suggestions and Questions
-----

* How were heat maps shown picked? Were the images and queries picked at random, and then the heat maps visualized? Or visualized first and then picked after?
* I think the attention mechanism (such as $\mathcal{A}_{IS}\(.\)$) used here isn't really an attention mechanism in the true sense. It seems a little closer to gating the different image and text embeddings based on the modifier text than attending over them (which would use the modifier text _and_ the image, for example, to generate the attention maps).

Minor comments
-----

* Page 2: "conciliates" -> "conciliate" (the word itself sounds incorrect in this context though, did you mean combine?)

**Summary Of The Paper:**

This paper focuses on the task of image retrieval using a sample image combined with text modifiers. It does this by the use of two modules that compute the implicit similarity (image-image similarity) and explicit matching (image-text similarity) scores. The modules computing both these scores use attention mechanisms so as to be conditioned on the text modifier. The scores are summed for the final retrieval. The efficiency of this architecture is shown quite clearly by comparing it with other state-of-the-art architectures on 3 different datasets. Ablations and qualitative analysis further highlights the utility of the architecture and its different components.



**Summary Of The Review:**

Based on all the strengths (extremely well written and well motivated; clear; a simple but novel architecture; comprehensive experiments and good ablations; great results), I'd recommend this paper be accepted.

---

> ### Author Response · Authors · 2021-11-17
> **Response to RL8FA**
>
> We would like to thank the reviewer for an insightful review. We are glad the reviewer found the paper “extremely well written and well motivated”, the idea “novel”, and the experiments “comprehensive”.
> We answer questions and concerns from the reviewer below.
>
> > How were heat maps shown picked? At random or visualized first and then picked after?
>
> The process to choose the heatmaps reported in Figure 3, 11, and 12 is as follows. We randomly selected queries for which the correct target is ranked first (as we openly mention in the captions of these three figures). We made this choice as we expect the heatmaps of those queries to be more interpretable. Our qualitative results cover a broad range of examples (including the 3 categories from FashionIQ, and Shoes). We are happy to add totally random examples or failure cases in the supplementary if the reviewer thinks it would provide more insights. Yet, our experience is that heatmaps of such failing queries are hard to interpret.
>
> > The attention mechanism used here isn’t really an attention mechanism in the true sense. It is closer to gating the image and text embeddings than attending over them.
>
> We totally agree that $\mathcal A_{IS}$ and $\mathcal A_{EM}$ fundamentally differ from attention mechanisms as defined in [A], which has recently become the most standard definition of an attention mechanism.
> However, *attention* is a term that has been heavily used in literature in the past. It has been used for instance to refer to a weight map in [B], a weighting vector with a fully-connected layer in [C] (which *Reviewer pgma* pointed us to), to a weight map based on entropy in [D], and even averages over the channels of the feature map in [E]. This non-exhaustive set of examples illustrates a less constrained use of the term “attention”, and we would like to keep this term that, we think, intuitively conveys the fact that ARTEMIS reweights (focusing its attention on) some of the visual and textual feature dimensions. Yet, following the reviewer’s valid point, we have clarified this by adding a dedicated paragraph at the end of the related work section (Section 2).
>
> [A] Vaswani et al., in the paper “Attention is all you need” (NeurIPS 2017).
>
> [B] Xu et al., Show, Attend and Tell: Neural Image Caption Generation with Visual Attention (ICML’15)
>
> [C]  Li et al, Person Search with Natural Language Description (CVPR 2017)
>
> [D] Wang et al., Transferable Attention for Domain Adaptation (AAAI’19)
>
> [E]  Kang et al., Deep Adversarial Attention Alignment for Unsupervised Domain Adaptation: the Benefit of Target Expectation Maximization (ECCV 18)
>
> > “Page 2: "conciliates" -> "combine”
>
> Good catch, thank you. We have fixed this.

---

> > ### Comment · Reviewer_L8FA · 2021-11-23
> > **Reply to author response**
> >
> > I would like to thank the authors for their response and for the clarifications and the updated paper. Overall, I continue to maintain that this is a good paper, and keep my rating of an 8 (accept).

---

### Official Review · Reviewer_5uBZ · 2021-11-03

**Correctness:** 4
**Technical Novelty And Significance:** 2
**Empirical Novelty And Significance:** 2
**Recommendation:** 8
**Confidence:** 4

**Main Review:**

**Strengths**
- The motivation and exposition of the problem and the solution is very clear.
- The paper proposes simple yet effective and practically applicable method.
- The experimental evaluation is solid, ablation experiments are through and limitations are discussed, albeit in supplementary material

**Weaknesses**
- Not sure how much the improvement is significant, if results are reported as average over three runs why there is no variance?
- Explicit complexity analysis would help understand the advantage of the method against more computationally complex (e.g. cross-attention) methods.
- “We select the best performing model on the validation set to report results on the test set and vice-versa” is not comparable to other reported methods since they didn’t have access to test set. The comparison with TIRG is valid, assuming that the protocol was the same for both ATERMIS and TIRG.

**Summary Of The Paper:**

The paper presents a method for the task special kind of multi-modal retrieval: image search with free-form text modifiers, where image is used as a query and accompanied text specifies the differences with respect to the given query image which target image should satisfy. It is practically important problem for which several benchmark datasets exist. The paper proposes to use the representation of text modifier to attend (modulate) both to query and target image representations, and to use these modulated representations to obtain image and text matching scores, which are then averaged to obtain final score for each target image, as displayed in Figure 2. The proposed model architecture is conceptually simple and lightweight, allowing thus scalable retrieval system, while outperforming several recent baselines on three challenging benchmarks: Fashion IQ, Shoes and CIRR dataset. The paper also presents qualitative results of attention via Grad-CAM.


**Summary Of The Review:**

The paper presents a simple method for important problem that outperforms recently proposed methods on three challenging benchmarks. The paper is clearly written, backed with experimental evaluation and ablations, where it shows good performance.

---

> ### Author Response · Authors · 2021-11-17
> **Response to R5uBZ**
>
> We would like to thank the reviewer for an insightful review. We are glad that the reviewer found “the motivation and exposition of the problem and the solution very clear”, and the method “simple yet effective and practically applicable”.
> We answer questions and concerns from the reviewer below.
>
> > Not sure [..] improvement is significant [..] results are reported as average over three runs, why there is no variance?
>
> We updated Tables 2 to 5 from our submission (and Table 6 from the Appendix) to add the standard deviations for the experiments we ran ourselves (ARTEMIS and TIRG) over the three runs.  We note that, in general, the standard deviation of ARTEMIS’ results is smaller than the difference between the overall first and second results (marked in **bold** and **blue**, respectively).
>
> > Explicit complexity analysis would help understand the advantage of the method against more computationally complex (e.g. cross-attention) methods.
>
> Thanks for the great suggestion. We agree that this study would ground some of the claims we make in the paper. We have added an entire section about complexity in the appendix (see section C) including a quantitative complexity comparison in Table 7 which provides evidence of ARTEMIS’s lower computational complexity compared to other approaches such as CoSMo and VAL.
>
>
> > The best performing model on the validation set is reported on the test set and vice-versa, so this is not comparable to other methods, since they didn’t have access to the test set
>
> Although our protocol is indeed slightly different, we argue it does not provide any advantage to our method, and is even more principled as:
>
> - our FashionIQ models are trained on the same data as the other methods reported in Table 2
> - the test split is only used to choose a checkpoint for the val set (and vice versa)  which is standard for train-val-test datasets. In contrast, since other approaches did not have access to the three splits, they most likely used the last checkpoint by default, and hence may have resorted to ad-hoc strategies to select the number of training epochs.
>
> In order to fully validate the claim that our protocol does not offer an unfair advantage, we also report below the results of ARTEMIS (RN50 + BiGRU) on FashionIQ, on the test and val splits, when we evaluate on the last checkpoint ('last ckpt', protocol of previous works), and compare them with results obtained with our protocol ('xValidation', presented in Table 2 & 3). We observe that this incurs very little change, and that all our conclusions from Table 2 & 3 still hold.
>
> |                          |  	CM 	|  Dress R@10 |  Shirt R@10 | Toptee R@10 |  Mean R@10  |  Dress R@50 |  Shirt R@50 | Toptee R@50 |  Mean R@50  |
> |----------------------------------|:-----------:|:-----------:|:-----------:|:-----------:|:-----------:|:-----------:|:-----------:|:-----------:|:-----------:|
> |                              	|         	|         	|         	|         	|         	|         	|         	|         	|         	|
> | ARTEMIS, val split, xValidation  | 37.68±0.15  | 25.68±0.53  | 21.57±0.86  | 28.59±0.49  | 25.28±0.40  | 51.05±0.34  | 44.13±0.38  | 55.06±0.61  | 50.08±0.18  |
> | ARTEMIS, val split, last ckpt| 37.69±0.15  | 25.76±0.18  | 21.54±0.93  | 28.59±0.53  | 25.30±0.25  | 51.06±0.04  | 44.08±0.58  | 55.09±0.43  | 50.08±0.18  |
> | ARTEMIS, test split, xValidation | 36.63±0.22  | 27.58±0.33  | 20.92±0.09  | 25.65±0.88  | 24.72±0.43  | 50.93±0.26  | 44.40±0.71  | 50.32±0.42  | 48.55±0.08  |
> | ARTEMIS, test split, last ckpt   | 36.56±0.06  | 27.49±0.12  | 20.95±0.01  | 25.52±0.94  | 24.66±0.21  | 50.88±0.07  | 44.34±0.40  | 50.20±0.34  | 48.47±0.00  |
>
> > The comparison with TIRG is valid, assuming that the protocol was the same for both ARTEMIS and TIRG
>
> We confirm that our re-implementation of TIRG followed the exact same protocol as ARTEMIS (see also first paragraph of section 4.3).

---

> > ### Comment · Reviewer_5uBZ · 2021-11-30
> > **Comment on the authors’ response**
> >
> > Thank the authors for addressing the concerns in revised version of the paper. They have only reinforced my initial assessment and I therefore keep the score.

---

### Author Response · Authors · 2021-11-17
**Official Comment on Rebuttal Revision**

We thank the three reviewers for their careful read of our paper and their insightful feedback. We are glad that they found the paper to be clearly written (all three reviewers), to tackle a practically important problem (*R5uBZ*) with a simple (all three reviewers), novel (*RL8FA*), and effective (*R5uBZ, Rpgma*) approach, and to propose a solid experimental evaluation (all three reviewers).

We provide individual responses to their questions and comments below.

Note that we have also uploaded an updated version of our manuscript which mostly features the following changes:
- We have updated Table 2 to 6, to add the standard deviation to our quantitative results, following *R5uBZ*’s question.
- We have slightly extended the related work section, following *Rpgma*’s suggestion.
- We have added a whole new section in the supplementary material (section C) about complexity and efficiency, a common request of *R5uBZ* and *Rpgma*. It clearly shows the advantage of our method against CoSMo (Lee et al., 2021).

---

### Decision · Program_Chairs · 2022-01-20

**Decision:**

Accept (Poster)

**Comment:**

Reviewers viewed the proposed approach to image retrieval as both simple and effective, the manuscript as well written and well motivated, and the presented experiments as relatively compressive (spanning three datasets). There was some discussion about novelty -- all reviewers viewed the approach as simple, but one had concerns about the approach's novelty relative to past work. This concern, as well as some concerns about experimental evaluation, were adequately addressed in author response.